# Proteomic Analysis of *Antheraea pernyi* Larvae After Injection of Vitamin C and Preliminary Exploration of the Function of *Antheraea pernyi* Aldonolactonase

**DOI:** 10.3390/biology14091236

**Published:** 2025-09-10

**Authors:** Xin Xu, Zhongwen Liu, Yongjun Zhang, Yaoting Zhang, Yanxian Lian, Zhen Zhang, Xuwei Zhu

**Affiliations:** 1Henan Sericultural Research Institute, Zhengzhou 450008, China; xuxinsdau@163.com (X.X.); 18567289660@163.com (Z.L.); zyj19680505@163.com (Y.Z.); ckyyaoting@126.com (Y.Z.); 2College of Food and Bioengineering, Henan University of Animal Husbandry and Economy, Zhengzhou 450008, China; lianyanxian@hnuahe.edu.cn

**Keywords:** *Antheraea pernyi*, Vitamin C, aldose lipase, proteome

## Abstract

VC (Vitamin C) is an essential substance for regulating the physiological functions of *Antheraea pernyi* and an important component in nutrients and artificial feed. We analyzed the midgut proteome of larvae injected with VC and used GO (Gene Ontology) functional annotation and KEGG (Kyoto Encyclopedia of Genes and Genomes) analysis to identify proteins related to the metabolism of VC in *A. pernyi*. Then, we cloned the ApALase (Antheraea pernyi Aldonolactonase) gene and performed bioinformatic analysis. RT-qPCR was used to determine the effect of different VC injection doses on the relative expression level of *ApALase* in fourth instar larvae at 24 h. Preliminary testing of the gene’s function is expected to assist in studying the molecular mechanism of VC metabolism in *A. pernyi*.

## 1. Introduction

*Antheraea pernyi* is a large silk-producing insect belonging to the family Saturniidae of the order Lepidoptera. Tussah leaves are the primary feed for the rearing of *A. pernyi*, and the various nutrients in oak leaves can meet their growth and development needs, of which VC is an indispensable substance for regulating physiological functions. VC regulates the physiological functions of *A. pernyi*, participates in amino acid and lipid metabolism, and enhances the body’s immunity. At the same time, VC is an important component in nutrients and artificial feed during the rearing process. VC is an important cellular factor in living organisms [1]. It participates in amino acid metabolism, the synthesis of neurotransmitters, and the synthesis of collagen and intercellular matrix while preventing the synthesis of nitrosamines, stimulating the phagocytic activity of leukocytes, promoting the formation of antibodies, reducing capillary permeability, stimulating blood coagulation, increasing resistance to infections, preventing cancer, and enhancing immune function. It can also cause lipid and protein oxidation and DNA damage [2].

Insects contain higher levels of vitamins such as A, B1, B2, D, E, and C [3]. Mineral elements and vitamins are indispensable substances for maintaining normal physiological metabolism in insects, and their content varies with the developmental process to meet the physiological needs of insect growth and development at different stages [4]. The VC content in the yellow-brown field cricket gradually increases from the egg stage to the seventh instar nymph stage, then gradually decreases from the ninth instar nymph stage to the 3-day-old adult stage [5]. Agata et al. have proven that VC is an important and essential nutrient for *Bombyx mori* [6,7]. Research also indicates that VC is one of the most significant feeding stimulants for *Bombyx mori* [8]. However, when measuring the electrophysiological responses of silkworms’ taste to VC, Cui Weizheng et al. found that VC, sucrose, and inositol can mutually inhibit discharge pulses in taste receptors. Therefore, the relationship between VC and the feeding habits of silkworms requires further study [9].

Recent research suggests that the VC synthesis pathway in animals is the L-gulonic acid pathway, which includes five steps [10]. The enzyme involved in the penultimate step of this synthesis is ALase (aldonolactonase). ALase is a multifunctional protein with the functions of SMP30 (senescence marker protein-30), GNL (gluconolactonase), RGN (regucalcin), and LRE (luciferin regenerating enzyme-like region family protein) [11]. It is involved in the homeostasis of intracellular calcium ions (Ca^2+^), participates in VC synthesis in non-primate mammals, and is highly conserved in vertebrates. The enzymatic activities of ALase include aldonolactonase and organophosphate hydrolysis, and it has anti-apoptotic and oxidative stress resistance properties [11]. However, despite it being a key enzyme in the VC metabolic pathway, there have been no related studies on ALase in *A. pernyi* to date, and research on its metabolic pathway remains lacking.

## 2. Materials and Methods

### 2.1. Experimental Materials

The variety of *A. pernyi* used was Yuda No.1, bred and provided by the Henan Academy of Sericulture Science.

### 2.2. Proteomic Analysis of Midgut Proteins in A. pernyi Larvae After VC Injection

Based on preliminary experiments, 4th instar *A. pernyi* larvae at 24 h were selected for VC injection. The injection dose was 0.02 mg/g body weight, while the control group received an equal volume of 0.9% saline. Midguts from 10 larvae were collected for each sample, with 3 replicates per group. Samples were immediately frozen in liquid nitrogen and stored at −80 °C for further use.

#### 2.2.1. Protein Extraction

We weighed an appropriate amount of sample; transferred it into a 2 mL centrifuge tube; added two steel balls; added an appropriate amount of 1XCocktail containing SDS L3 and EDTA(Ethylenediaminetetraacetic Acid); placed it on ice for 5 min; and added DTT for a final concentration of 10 mM. We used a grinder (BJ-400A, Deqing Baijie Household Products Trading Co., Ltd, Huzhou, China) (60 Hz frequency, 2 min) for crushing and cracking, and the sample was centrifuged at 25,000× *g* and 4 °C for 15 min. We then took the supernatant. We added DTT (Dithiothreitol) with a final concentration of 10 mM and immersed it in a water bath at 56 °C for 1 h. We added IAM(Iodoacetamide) with a final concentration of 55 mM and placed it in a darkroom for 45 min. Then, we added cold acetone to the protein solution in a 1:5 ratio and refrigerated it at −20 °C for 30 min. We then centrifuged it at 25,000× *g* and 4 °C for 15 min and discarded the supernatant. Then, we air-dried and precipitated the sample, added an appropriate amount of SDS (sodium dodecyl sulfate)-free L3, and used a grinder (60 Hz frequency, 2 min) to promote protein dissolution. Then, we centrifuged the sample at 25,000× *g* and 4 °C for 15 min to obtain the supernatant, which was the protein solution.

#### 2.2.2. Proteolysis After Quality Control of Protein Extraction

We added 100 μg of the hydrolyzed protein (adjusted according to project requirements) to a 1.5 mL centrifuge tube. If there was a high concentration of urea or SDS in the protein solution, it was diluted with 0.5 M TEAB (tetraethylammonium bromide) to make a final urea concentration of less than 2 M. The final SDS concentration was less than 0.1%. We added Trypsin enzyme (μg) to the enzyme solution for a Trypsin–substrate protein (μg) ratio of 1:20, centrifuged it at low speed for 1 min, and incubated it at 37 °C for 4 h. Then, we took out the digested peptide solution for desalination operation and froze and dried the peptide extract obtained after desalination.

#### 2.2.3. High-Performance Liquid Chromatography

We dissolved the dried peptide sample in mobile phase A (2% ACN (Acrylonitrile); 0.1% FA (Fulvic Acid)) and centrifuged it at 20,000× *g* for 10 min, taking the supernatant for injection. We used Thermo UltiMate 3000 UHPLC (Thermo Fisher Scientific, Waltham, MA, USA) (ultra-high-performance liquid chromatography) for separation. We placed the sample in a trap column for enrichment and desalination, connected in a series with a self-loaded C18 column (75 μm inner diameter, 3 μm column particle size, 25 μm column length); then, it was separated at a flow rate of 300 mL/min through the following effective gradient: 0–5 min, 5% mobile phase B (98% ACN; 0.1% FA). From 5 to 45 min, we increased mobile phase B linearly from 5% to 25%, and after 45–50 min, we increased mobile phase B from 25% to 35%. After 50~52 min, we increased mobile phase B from 35% to 80%. After 52~54 min, we added 80% of mobile phase B. After 54–60 min, we added 5% of mobile phase B. At the end of the process, liquid phase separation was directly viewed under a mass spectrometer.

#### 2.2.4. Mass Spectrometry Detection

The peptide segments separated during the liquid phase were placed in an ESI (Electrospray Ionization) tandem mass spectrometer (TripleTOF 5600; SCIEX, Framingham, MA, USA). The ion source was a Nanospray III source (SCIEX, Framingham, MA, USA), and the emitter was a quartz material drawn needle (New Objectives, Woburn, MA, USA). During data acquisition, the parameters of the mass spectrometer were set as follows: the spray voltage of the ion source was 2300 V, the nitrogen pressure was 35 psi, the spray gas was 15, and the temperature at the spray interface was 150 °C. Scanning was performed in high-sensitivity mode, with a cumulative time of 250 ms for primary mass spectrometry scanning and a scanning mass range of 350–1500 *m*/*z*. Based on the first-level scanning information, from highest to lowest, we selected the top 40 ions with intensities exceeding 150 cp in the first-level spectrum for fragmentation and scanned the second-level information. The screening criteria were as follows: (1) the *m*/*z* range was 350–1250; (2) the number of charges was 2–5; and (3) the dynamic exclusion of parent ions was set to ensure that the fragmentation of the same parent ion did not exceed half of the peak time (approximately 12 s). The scanning accumulation time of the secondary mass spectrometry was 50 ms. We chose the “Rolling Collision Energy” setting for fragmentation energy.

#### 2.2.5. Protein Identification

The offline data was identified using the Andromeda engine integrated with MaxQuant. At the spectral level, it was filtered at a PSM-level FDR ≤ 1%, and at the protein level, it was further filtered with a protein-level FDR ≤ 1% to obtain significant identification results.

#### 2.2.6. MaxQuant Parameter Configuration

MaxQuant (http://www.maxquant.org accessed on 26 August 2025) is a free protein identification and quantification program developed by the Max Planck Institutes in Germany, suitable for high-precision mass spectrometry data. The version used in this project was MaxQuant 1.5.3.30. We used the original offline data as the input file, configured the corresponding parameters and database, and then performed identification and quantitative analysis.

MaxQuant parameter configuration: MaxQuant 1.5.3.30, enzyme (Trypsin), Peptide_Mass_Tolerance (4.5 ppm), Fragment_Mass_Tolerance (20 ppm), minimal peptide length (7), PSM-level FDR (0.01), protein-level FDR (0.01), fixed modifications (Carbamidomethyl (C)), variable modifications (Oxidation (M), Acetyl (Protein N-term), Deamidated(NQ), Gln->pyro-Glu)), and Database (sequence_filtered.fasta (4252 sequences)).

Based on the GO functional annotation results and official classification, the differentially expressed proteins were classified into functional categories. The phyper function in R was used for enrichment analysis, and the *p*-value was calculated. Then, FDR correction was performed on the *p*-value, also known as the *Q*-value. Functions with *Q* ≤ 0.05 were usually considered significantly enriched, and the top 20 GO functional items with the smallest *Q*-values were plotted. According to the KEGG annotation results and official classification, the differentially expressed proteins were classified into biological pathways, and the rest underwent GO functional annotation.

#### 2.2.7. Protein Quantification and Differential Analysis

This process used MaxQuant to extract peak area intensity values and calculate protein quantification values. Then, according to the set comparison group, we calculated the fold difference in proteins in each comparison group and performed significance testing using Welch’s *t*-test. Furthermore, a difference factor > 1.5 and a *p*-value < 0.05 were the criteria for significant differences in screening. Finally, enrichment analysis was performed on the differentially expressed proteins.

### 2.3. Cloning and Bioinformatic Analysis of ApALase Gene

Based on the analysis of differentially expressed proteins of the proteome described in Section 3.2, conservative structural regions were selected. Combined with PCR primer design principles, Primer Premier 5.0 was used to design *ApALase*-specific primers (Table 1), and *ApA* (*Antheraea pernyi Actinoin*) was selected as the internal reference gene (Table 1). The annealing temperature was 50 °C with 30 cycles.

According to its instructions, we used an animal tissue total RNA extraction kit (RNAprep Pure Tissue Kit) to extract the total RNA from *A. pernyi* larvae infected with microsporidia. We reversed the RNA into cDNA using the FastQuant cDNA first-strand synthesis kit, according to its instructions. PCR reaction system: 2 × Taq PCR Mastermix 12.5 μL; 1 μL each of forward and reverse primers (10 μ mol/L); and 1 μL of cDNA and 25 μL of ddH_2_O added. Reaction procedure: pre-denaturation at 94 °C for 10 min; denaturation at 94 °C for 30 s; annealing at 50 °C for 30 s; and extension at 72 °C for 30 s for a total of 35 cycles. This was followed by an extension at 72 °C for another 10 min. After the reaction, we used 1.0% agarose gel electrophoresis to detect the PCR product; recovered the target fragment; connected the recovered product to a pMD18-T carrier overnight at 16 °C; transformed the connecting product into *Escherichia coli* DH5α; screened blue and white spots on the medium containing ampicillin; randomly selected white colonies for colony identification; and screened out positive clones, which were then sequenced and identified by Shanghai Sangong Biotechnology Technology Service Co., Ltd. (Shanghai, China).

The ApALase signal peptide and transmembrane region analysis of *A. pernyi* was performed using the online software SMART (https://smart.embl.de accessed on 26 August 2025, and the secondary structure was predicted online using the PSIBED website (https://bioinf.cs.ucl.ac.uk/psipred/ accessed on 26 August 2025). Protein advanced structure models were predicted using the online software SWISS-MODEL (https://swissmodel.expasy.org accessed on 26 August 2025). The bioinformatic prediction of the ApALase protein in *A. pernyi* was based on RGN and SWI/SNF complex subunits. To investigate its evolutionary relationship with similar proteins in different insects, this study used the Silkworm Genome Database and NCBI to select SWI/SNF complex subunit sequences from different insects. Multiple sequence alignment was performed on the ApALase amino acid sequence in the oak silkworm using MEGA 11.0, and Neighbor Joining was used to construct an evolutionary tree. The bootstrap replica value was 1000, and the mode was the Poisson model. Thus, the systematic evolutionary relationship was analyzed.

### 2.4. Effect of VC Injection on ApALase Gene Expression in Fourth A. pernyi Larvae

According to the preliminary test results, the VC injection dose gradients were 0, 0.01, and 0.02 mg/g body weight, and the 4th instar tussah was 24 h old. The mRNA in the midgut of the *A. pernyi* larvae was extracted at 6 time points after the injection: 0, 1, 6, 12, and 24 h. The RNA purity and content were determined by an ultra-micro UV spectrophotometer. All RNA samples were diluted to 1 μg/μL. The RNA was inverted into cDNA according to the instructions of the reverse transcription kit. PCR amplification was performed with the cDNA of each sample as the template. *ApA* (*Antheraea pernyi action*) was used as the positive control, and a differential expression analysis of the *ApALase* gene was performed.

Based on the ApALase sequence obtained from clone sequencing, conservative structural regions were selected, and RT-qPCR primer design principles were followed. Primer Premier 5.0 was used to design RT-qPCR-specific primers for *ApALase* (Table 1), with a target fragment length of 221 bp; *ApA* was the internal reference gene, and its qRT-PCR-specific primers were designed (Table 1) with a target fragment length of 134 bp. We performed qRT-PCR according to the experimental method of the FastFire rapid fluorescence quantitative PCR premix reagent (SYBR Green). Reaction program: 95 °C for 1 min; 95 °C for 5 s; 60 °C for 10 s; and 72 °C for 15 s, with a cycle count of 40.

### 2.5. Data Analysis

We performed relative gene expression analysis using the 2^−ΔΔCt^ method. GraphPad Prism 10 was used for data processing and plotting, and the SPSS 26.0 data analysis software was used for one-way ANOVA for significance analysis.

## 3. Results

### 3.1. Proteomic Data of Midgut Proteins in A. pernyi Larvae After VC Injection and GO Functional Annotation

The mass spectrometry instrument used in this study was the TripleTOF 5600. The raw data were processed using MaxQuant with the Andromeda search engine. Data were filtered at a PSM-level FDR ≤ 1% and further filtered at a protein-level FDR ≤ 1%. A total of 166 proteins and 1858 peptides were identified across all samples. The specific identification results for each sample are shown in the Table 2.

The following is a basic statistical chart of the protein identification results, described from three perspectives: “unique peptide distribution,” “protein mass distribution,” and “protein coverage distribution.”

We screened differential proteins between each group of samples and labeled the significance of the differences, setting boundary lines for differential protein screening and labeling the significance of the differences based on the scatter distribution of each protein. The statistical results for the differentially expressed proteins between the VC-injected silkworm larvae and the normal larvae showed that 4 proteins were significantly upregulated, and 1 protein was significantly downregulated, resulting in 5 differentially expressed proteins and 146 non-differentially expressed proteins (Figure 1).

After injecting VC into oak silkworm larvae, all identified proteins in the midgut proteome were subjected to GO functional annotation. In the biological process category, the largest number of proteins were annotated to “binding,” suggesting that this process exhibited a strong response in the midgut after VC injection. In the cellular component category, “cell” had the highest number of annotated proteins (70 in total), making it the most abundant classification. In the molecular function category, the largest number of proteins were associated with “cellular process” (Figure 2).

Since there were only five DEPs (differentially expressed proteins), no graphical representation was used for further analysis. Among them, two proteins were involved in catalytic activity, two were associated with binding, and one was linked to cells and cell parts. Although the number of DEPs was small, these five proteins are the key focus for functional cloning and analysis, particularly those related to catalytic activity, as they may include critical metabolic enzymes (Figure 3).

For proteins that are significantly downregulated, GO enrichment analysis provides GO entries for differentially enriched proteins, which often involve the biological functions that researchers are most concerned about. This result usually uses a *p*-value < 0.05 as a significantly enriched GO entry (Table 3.).

### 3.2. Functional Annotation and Subcellular Localization Analysis of the Midgut Proteome in A. pernyi Larvae Following VC Injection

Based on KEGG annotation, all identified proteins from the VC-treated and control groups were classified into six major functional branches: cellular processes, environmental information processing, genetic information processing, human diseases, metabolism, and organismal systems. The “translation” pathway contained the most proteins. Metabolism: The “global and overview maps” subcategory ranked second in protein abundance (Figure 4). Among the five differentially expressed proteins, two were annotated as “translation”; two belonged to “global and overview maps”; and the one remaining protein was distributed across other pathways (Figure 5). While GO analysis focuses on individual gene functions, KEGG emphasizes gene interactions within biological systems. The four proteins involved in translation and global metabolic pathways were prioritized for functional validation, gene interaction network analysis, and cloning and expression studies.

The following screenshot shows the pathway enrichment analysis webpage results. Pathways with *p*-values < 0.05 are significantly enriched in differentially expressed proteins. Due to the lack of an established protein library for tussah silkworms, the total number of proteins identified in the sample itself was relatively small. In addition, the entire experiment was divided into two large groups, resulting in significant individual differences in tussah silkworms and a relatively small number of differentially expressed proteins. Even if the screening conditions were set to the widest, this result would still be the same.

The KOGs (Eukaryotic Orthologous Groups) database is a commonly used protein function annotation database. Each KOG entry contains a series of orthologs or paralogs. Orthologous proteins refer to proteins that evolved from vertical lineages (speciation) of different species and typically retain the same function as the original protein. Parasitic homologous proteins are proteins derived from gene replication in a certain species, which may evolve new functions related to the original. Our KOG annotation analysis compared all identified proteins (BLAST 2.15.0) with the KOG database and obtained corresponding KOG annotation results, as shown in Figure 5 and Figure 6. By combining the KOG annotation analysis of all proteins and differentially expressed proteins, we found that translation, ribosomal structure, and biogenesis had the highest numbers, with 46 for all proteins and 2 for differentially expressed proteins (Figure 7). Combining GO analysis and KEGG pathway classification, we cloned and performed biological analysis on the differentially selected proteins.

Proteins were synthesized in ribosomes and transported by protein-sorting signals to specific organelles. Some proteins were secreted outside the cell or remained in the cytoplasm, and only by being transported to the correct location could they participate in various cellular life activities. The subcellular localization of proteins is an important part of protein function annotation. The WoLF PSORT II software was used to predict the subcellular localization of the proteins, and the specific analysis results are shown in Figure 8. Two proteins each were in the extra and cyto categories, and one was in the nucleus, indicating that most of the differentially expressed proteins were involved in the transport process.

### 3.3. Cloning Sequence and Biological Analysis of ApALase

Based on the analysis of differentially expressed proteins in the previous proteome, conservative structural regions were selected, and *ApALase*-specific primers were designed according to PCR primer design principles. The fragments were sequenced and found to have a total length of 709 bp, encoding 225 amino acids. The SMART program showed that ApALase is mainly composed of three structural domains, as shown in Figure 9A. The secondary structure prediction results for ApLase on the PSIBED website showed that there were 11 alpha helices and 3 beta folds in the sequence, as shown in Figure 9C. We submitted the ApaLise sequence to the SWISS-MODEL server to build a simulation structure, and the system’s QMEANscore4 score for this model was 0.42. There is a 20.25% similarity between ApALase and cytochrome C oxidase III in the simulated structure, as shown in Figure 9B.

ALase is a multifunctional protein that catalyzes the second-to-last step reaction in the de novo synthesis pathway of VC. It participates in the homeostasis of intracellular calcium divalent ions (Ca^2+^) and is involved in the synthesis of AsA in non-primate mammals. It is highly conserved in vertebrates and is predicted by bioinformatics to be a subunit of RGN and SWI/SNF complexes. To investigate the evolutionary relationship between the ApALase protein of *A. pernyi* and similar genes compared in BLAST, we identified 11 genes with similar sequences, including 8 belonging to the SWI/SNF complex subunit SMARCC2, indicating that ApALase should have a similar structure to the SWI/SNF complex subunit and is an aldose lipase (Figure 10).

### 3.4. The Effect of 24-h VC Injection on the Relative Expression Level of ApALase Gene in Fourth Instar A. pernyi

As shown in Figure 11, 1 h after VC injection, the expression levels of the treatment group were significantly reduced compared with the control group (*p* < 0.05). The decrease was more pronounced at high doses, indicating that in vitro VC injections can inhibit the expression of this gene. Moreover, the inhibitory effect persisted for 24 h compared with the 0.01 mg/g injection group and in the 0.02 mg/g injection group, where the gene expression level was significantly reduced compared with the control group (*p* < 0.01). This may be related to the slower VC metabolic cycle in which the enzyme participates.

## 4. Discussion

This study compared and analyzed the midgut proteome of tussah larvae before and after VC injection and identified and cloned the *ApALase* gene, which is related to the metabolism of VC in *A. pernyi*. The effect of injecting different doses of VC for 24 h at the fourth instar on the relative expression level of the *ApALase* gene was also determined. The differential proteomic data showed that catalytic activity is caused by a protein involved in the metabolism of key enzymes. The evolutionary results showed that *ApALase* has a similar structure to the SWI/SNF complex subunit and is an aldose lipase. Subsequent in vitro injection experiments preliminarily confirmed that this gene is related to the regulation of VC metabolism in *A. pernyi*.

## 5. Conclusions

Proteomics is a discipline that studies a complete set of proteins on a large scale and with high throughput. The development of mass spectrometry technology provides a more convenient method for discovering valuable proteins and has become an important differential protein-screening technique. This technology has also been applied to study the immune response of tussah microsporidia. Based on its horizontal transmission process [12], researchers have identified differential genes and proteins in the tussah midgut [13]. Subsequent studies have also found differentially expressed genes with significantly upregulated expression levels after infection [14].

VC is an antioxidant and coenzyme that is widely distributed in the animal and plant kingdoms. It is an essential nutrient for higher primates and many other organisms, and its nutritional effects have been proven in the growth and development of many herbivorous insects [15]. Pu Shenghou Xing et al. have shown that VC is an important essential nutrient for *Bombyx mori* [4,5,6] and can stimulate its feeding [16]. VC is an important component of some mulberry leaves and is an important additive in artificial feed for *Bombyx mori* [7]. As a unique biological resource in China, *A. pernyi* silk production is second only to that of the domestic silkworm. Although the various nutrients in the leaves of *A. pernyi* can meet the growth and development needs of the silkworm, research on adding nutrients to these leaves is of great significance. Through simple spraying, the nutritional value of *A. pernyi* leaves can be effectively improved, bringing certain economic benefits. In addition, compared with traditional feeding, artificial feeding with *A. pernyi* has obvious advantages, such as simplifying tussah production and significantly improving labor productivity. From these two research perspectives, VC is an essential substance for regulating the physiological functions of tussah silkworms and an important component in adding nutrients and artificial feed.

Alase has a total length of 307 amino acids and two structural domains: SGL and Cd, with a length of 245 amino acids. This region is present in *Bombyx mori* and various proteins expressed in eukaryotes and prokaryotes, including various enzymes, such as SMP30, GLN, and LRE. SMP30 hydrolyzes diisopropyl phosphate in the liver and shares sequence similarity with PON1 and LRE in this family. YvrR (Genebank accession number: COG3386) has a Cd length of 307 aa and is related to the transport and metabolism of carbohydrates [17]. The enzymatic activities of ALase include aldolase and phosphohydrolysis, as well as anti-apoptosis and oxidative stress [11]. Quantitative analysis has been conducted on the homologous genes *BmALase-like1* and *BmALase-like2* at different developmental stages and expression sites of *Bombyx mori* larvae, demonstrating that homologous genes in *Bombyx mori* are highly conserved [18]. Researchers have successively demonstrated that VC is an important essential nutrient for *Bombyx mori* [6]. It has been used as a substance with multiple functions, such as nutrition and stimulating feeding in the preparation of artificial feed, and the added amount far exceeds that of B vitamins, usually about 1% of the dry matter [7]. However, the evidence regarding the synthesis of VC in *Bombyx mori* is that the VC content increases before and after the reversal period during the development of eggs and embryos. From silk emergence to the pupal stage, the amount of VC in the body increases. On the second day of reduced pupa, the fat body can synthesize oxidized VC from mannose [13].

In the next step of this evolutionary research, we will continue to investigate the function of this gene by measuring the enzyme activity of key enzymes in the VC metabolic pathway under treatments with different concentrations of VC injection, validating its gene function from physiological and biochemical perspectives. If possible, we will conduct knockout experiments on this gene in oak silkworms to observe its function more closely.

## Figures and Tables

**Figure 1 biology-14-01236-f001:**
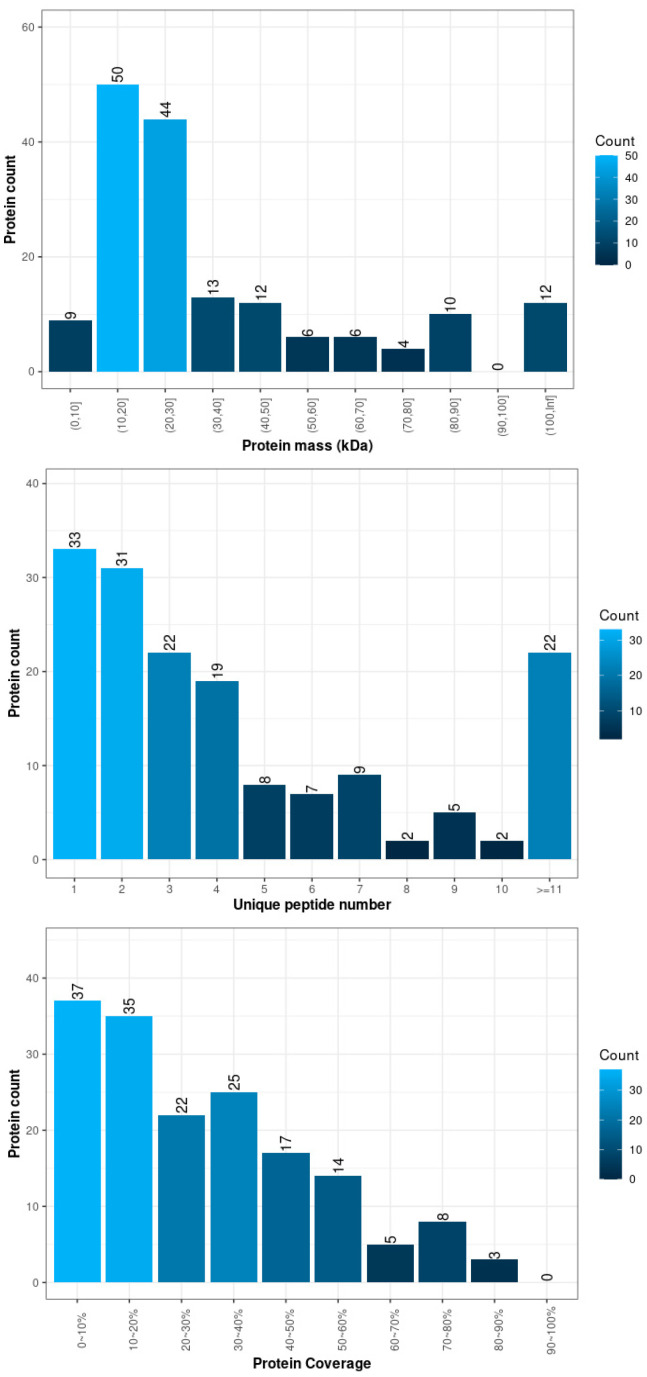
Basic statistical chart of protein identification results.

**Figure 2 biology-14-01236-f002:**
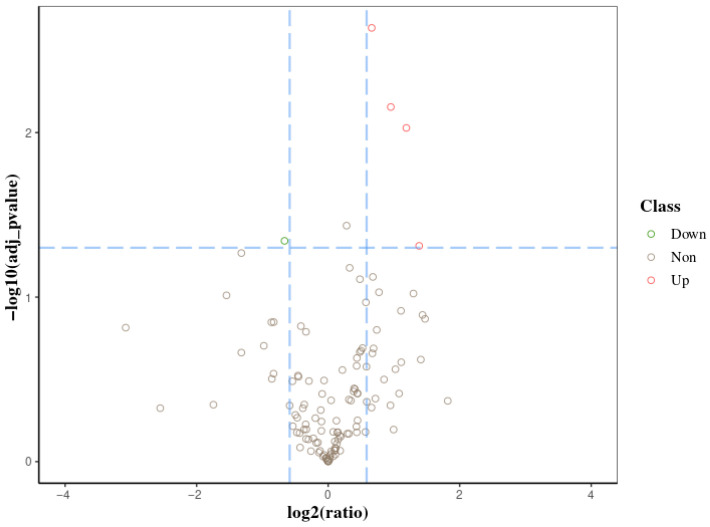
Volcano plot of differentially expressed proteins in the midgut proteome of *A. pernyi* larvae after VC injection.

**Figure 3 biology-14-01236-f003:**
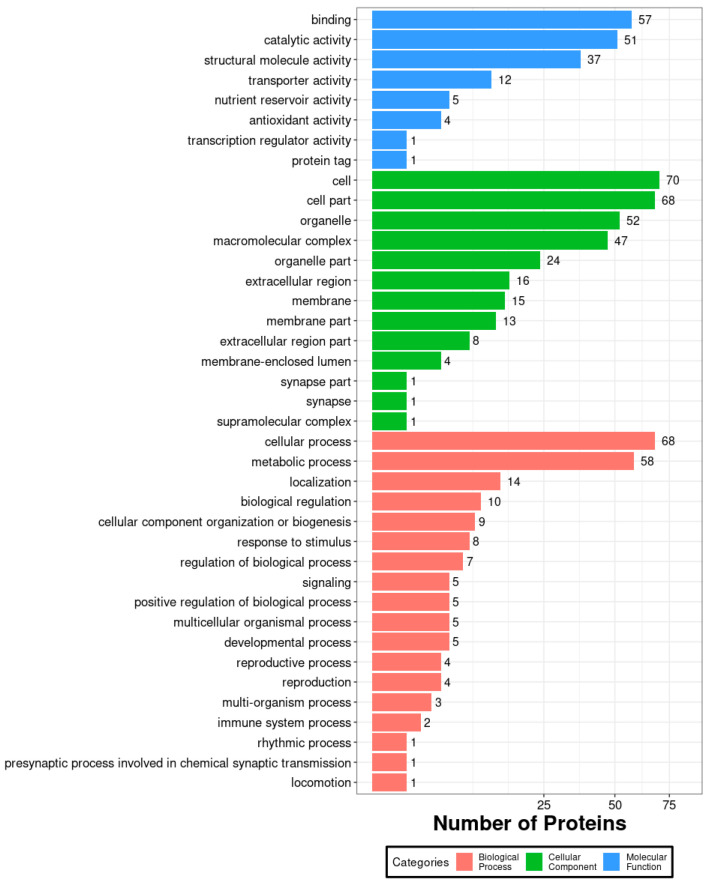
GO functional annotation of the midgut proteome in *A. pernyi* larvae after VC injection compared with the control group.

**Figure 4 biology-14-01236-f004:**
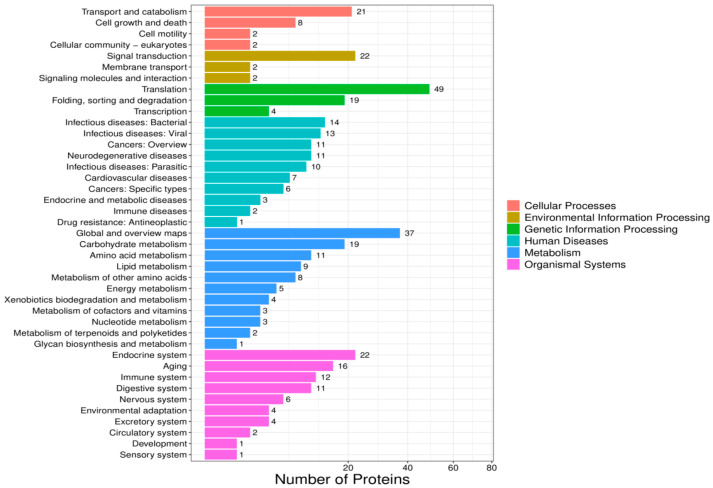
KEGG functional annotation of the midgut proteome in *A. pernyi* larvae after VC injection compared with the control group.

**Figure 5 biology-14-01236-f005:**
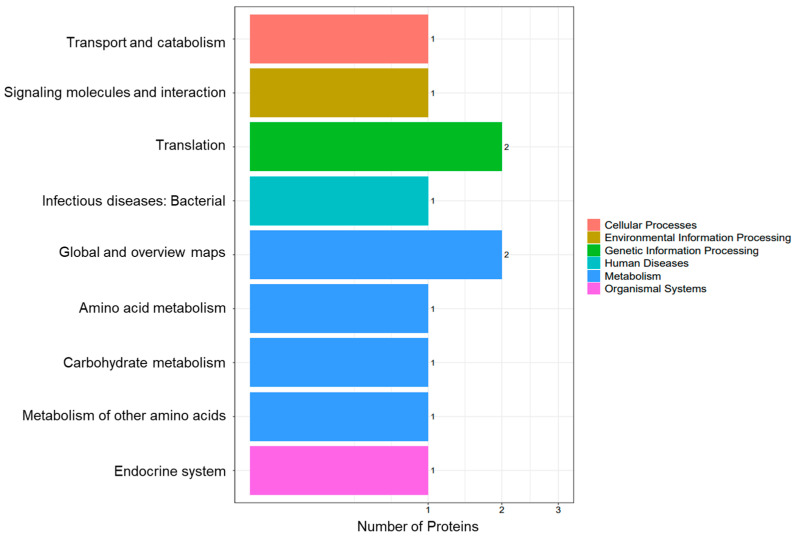
KEGG pathway classification of differentially expressed proteins between the midgut proteome and control group of *A. pernyi* larvae after VC injection.

**Figure 6 biology-14-01236-f006:**
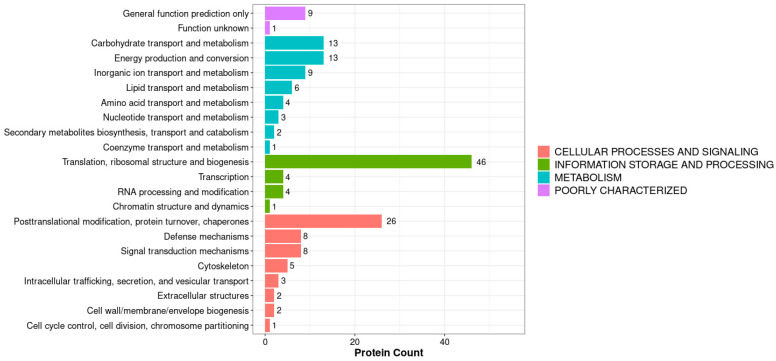
KOG annotation of all proteins in the midgut proteome of *A. pernyi* larvae after injection of VC compared with the control group.

**Figure 7 biology-14-01236-f007:**
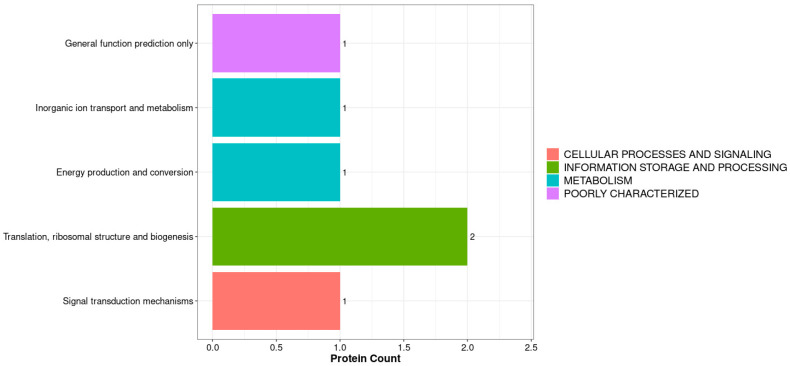
KOG Annotation of DEPs.

**Figure 8 biology-14-01236-f008:**
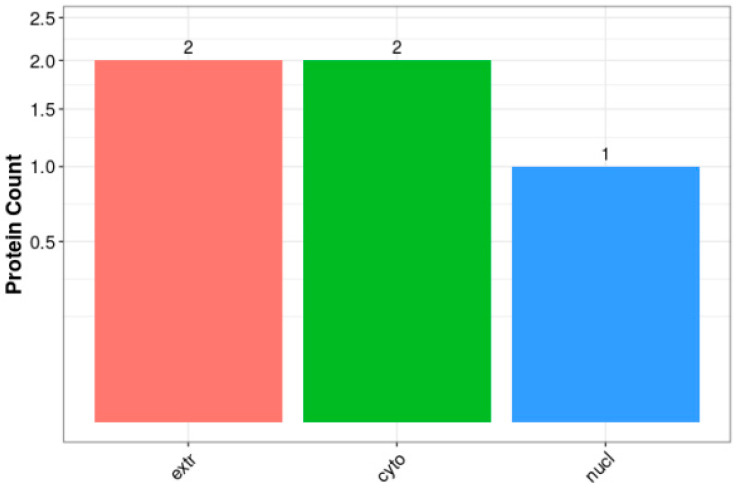
Histogram of subcellular localization of differential proteins.

**Figure 9 biology-14-01236-f009:**
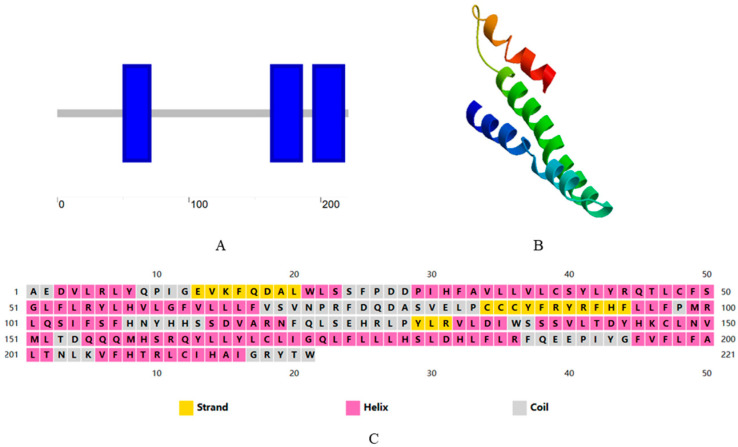
Prediction of amino acid sequence, secondary structure, and protein structure of *ApALase* from *A. pernyi.* (**A**–**C**) respectively, depict the simulation structure results for *ApALase* built on the SMART, PSIPRED, and SWISS-MODEL servers.

**Figure 10 biology-14-01236-f010:**
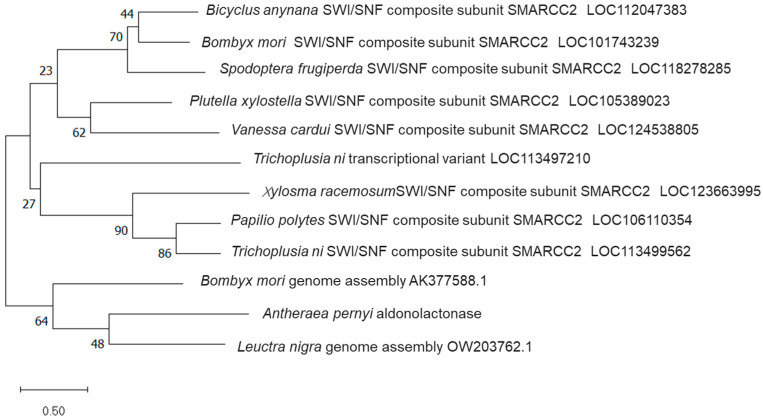
Phylogenetic tree of ApALase and similar proteins in different insects of *A. pernyi*.

**Figure 11 biology-14-01236-f011:**
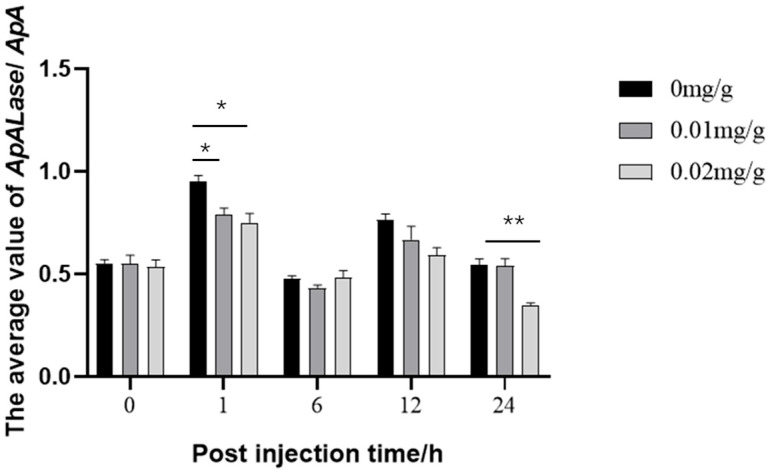
Changes in expression levels of *ApALase* during 24-h injection of different doses of VC into 4th instar larvae of *A. pernyi*. The symbol * indicates a significant difference from the control area, while ** indicates a highly significant difference from the control area.

**Table 1 biology-14-01236-t001:** Primer information.

Primers	Primer Sequences	Use
*ApALase*-F	TGTTTATGGACATGAAGCCG	Gene cloning
*ApALase*-R	CAGCATTATCGAATCGAAAGG	
*ApA*-F	CCAAAGGCCAACAGAGAGAAGA	
*ApA*-R	CAAGAATGAGGGCTGGAAGAGA	
q-*ApALase*-F	CCCATTGGAGAAGTGAAGT	qRT-PCR
q-*ApALase*-R	ACAAGAAGTGGAAGCGATA	
q-*ApA*-F	CGCTCCATCTACGATGAAGATC	
q-*ApA*-R	ACTCGTCGTACTCCTGTTTCG	

**Table 2 biology-14-01236-t002:** Specific identification results for each sample.

Sample	Total Spectra	Identified Spectra	Identified Peptides	Identified Proteins
LCK 1	28,367	3830	1284	149
LCK 2	28,147	3421	1255	146
LCK 3	27,179	3482	1279	146
LW 1	29,429	3402	1292	153
LW 2	28,071	3691	1324	152
LW 3	26,221	3475	1307	150

**Table 3 biology-14-01236-t003:** GO entries of the midgut proteome in *A. pernyi* larvae after VC injection compared with the control group.

Gene Ontology Term	Cluster Frequency	Protein Frequency of Use	*p*-Value
ribonucleoprotein complex	12 out of 36 genes, 33.3%	90 out of 766 genes, 11.7%	0.000389
intracellular	30 out of 36 genes, 83.3%	511 out of 766 genes, 66.7%	0.019471
chromatin	2 out of 36 genes, 5.6%	6 out of 766 genes, 0.8%	0.028608
intracellular part	29 out of 36 genes, 80.6%	503 out of 766 genes, 65.7%	0.036246
macromolecular complex	14 out of 36 genes, 38.9%	197 out of 766 genes, 25.7%	0.052778
cell	32 out of 36 genes, 88.9%	592 out of 766 genes, 77.3%	0.059895
non-membrane-bounded organelle	5 out of 36 genes, 13.9%	47 out of 766 genes, 6.1%	0.062586
endoplasmic reticulum	2 out of 36 genes, 5.6%	9 out of 766 genes, 1.2%	0.062811
chromosome	2 out of 36 genes, 5.6%	11 out of 766 genes, 1.4%	0.090476
cytoplasmic part	5 out of 36 genes, 13.9%	80 out of 766 genes, 10.4%	0.319755
cytoplasm	5 out of 36 genes, 13.9%	81 out of 766 genes, 10.6%	0.32958

## Data Availability

Data is contained within the article.

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
