# Peer review of "Proteomic Analysis of Antheraea pernyi Larvae After Injection of Vitamin C and Preliminary Exploration of the Function of Antheraea pernyi Aldonolactonase"

_biology, 2025, doi:10.3390/biology14091236_

Round 1
Reviewer 1 Report
Comments and Suggestions for Authors
In this paper, the influence of vitamin C injection on proteomics of Antheraea pernyi larvae was estimated. The topic looks quite interesting and scientifically sound; modern techniques, includiung proteomics, gene sequencing and quantitative PCR were used in the study. At the same time, the manuscript text should be significantly improved before acceptance.
Major comments:
- In 'Simple Summary' and 'Abstract' the authors should explain more clearly the importance of the study and why ApALase was chosen for sequencing and gene function analysis. Moreover, in 'Abstract' I noticed that ApALase gene cloning mentioned twice, so one of these sentences should be removed.
- In 'Introduction', more information about scientifical and applied meaning of Antheraea pernyi is needed. Also, from my point of view, the sequence of paragraphs should be changed. First, the description of the species should be made, then the authors can explain the meaning of VC metabolism, why proteomic analysis is important and its applied significance. The current version of Introduction looks quite incoherent.
- There is no need list all the equipment and reagents (Section 2.2.), because a paper is not a dissertation or master thesis. At the same time, some important information is missing, e.g. what method and evolutionary model were used to construct phylogenetic tree? Was bootstraping applied?
- Describing the qPCR results (3.4.), the authors should mention the fold changes in ApALase expression in injected samples in comrarison to control.
- In 'Discussion', the authors should discuss the obtained results and compare them to previous studies, not describing proteomics and VC significance. At least, some text from 'Conclusions' can be moved to 'Discussion'. I believe that much attention in this chapter must be paid on perspectives of this work and its applied significance.
Minor comments:
- I believe that in the title should be 'Vitamin C' instead of 'VC' (these abbreviation must be explained below).
- Line 16: 'aldolactonase' should not be in Italics.
- Lines 58-59: The sentence 'A. pernyi is a large silk-producing insect.....' looks strange in this paragraph.
- Line 60-63: This sentence should be divided or changed.
- Line 99: Why this amount of VC was injected?
- Line 169: What RNA extraction kit was used?
- Line 198: What equipment was used to estimate RNA concentration and purity? If it was specrophotometer, then it is not correct to say that concentration was measutred by determining OD260/280. This ration only demonstrates nucleic acids purity from protein contamination.
- Figure 10. The meaning of asterisks should be explained.
- Figure 10. In this figure, two concentrations (0.01 and 0.02) are compared, although only 0.02 mg/g was mentioned in the text.
Author Response
Major Comments 1: [1. In 'Simple Summary' and 'Abstract' the authors should explain more clearly the importance of the study and why ApALase was chosen for sequencing and gene function analysis. Moreover, in 'Abstract' I noticed that ApALase gene cloning mentioned twice, so one of these sentences should be removed.]
Response 1: [Thank you for your correction. The article has been revised accordingly.]
Major Comments 2: [In 'Introduction', more information about scientifical and applied meaning of Antheraea pernyi is needed. Also, from my point of view, the sequence of paragraphs should be changed. First, the description of the species should be made, then the authors can explain the meaning of VC metabolism, why proteomic analysis is important and its applied significance. The current version of Introduction looks quite incoherent.]
Response 2: [Thank you for your correction. The order of the paragraphs in the introduction has been adjusted]
Major Comments 3: [There is no need list all the equipment and reagents (Section 2.2.), because a paper is not a dissertation or master thesis. At the same time, some important information is missing, e.g. what method and evolutionary model were used to construct phylogenetic tree? Was bootstraping applied?]
Response 3: [The content of 2.2 has been deleted.Thank you for your correction. We had added “the method for constructing an evolutionary tree is Neighbor Joining, and the value of Bootstrap replicas is 1000, and the Mode is the Poisson model”in the Section 2.3.]
Major Comments 4: [Describing the qPCR results (3.4.), the authors should mention the fold changes in ApALase expression in injected samples in comrarison to control.]
Response 4: [Thank you for your reminder. We have added a description of the concentration gradient in the result analysis.]
Major Comments 5: [In 'Discussion', the authors should discuss the obtained results and compare them to previous studies, not describing proteomics and VC significance. At least, some text from 'Conclusions' can be moved to 'Discussion'. I believe that much attention in this chapter must be paid on perspectives of this work and its applied significance.]
Response 5: [Thank you for your question. The article has been revised accordingly.]
Minor Comments 1: [1. I believe that in the title should be 'Vitamin C' instead of 'VC' (these abbreviation must be explained below).]
Response 1: [Thank you for your correction. The article has been revised accordingly.]
Minor Comments 2: [Line 16: 'aldolactonase' should not be in Italics.]
Response 2: [Thank you for your correction. The article has been revised.]
Minor Comments 3: [Lines 58-59: The sentence 'A. pernyi is a large silk-producing insect.....' looks strange in this paragraph.]
Response 3: [This sentence has been deleted.]
Minor Comments 4: [Line 60-63: This sentence should be divided or changed.]
Response 4: [The paragraph has been modified]
Minor Comments 5: [Line 99: Why this amount of VC was injected?]
Response 5: [The preliminary experiments of the research group showed that the injection dose per gram of body mass of silkworms is 0.02 mg can promote the growth and development of silkworm larvae, improve resistance, increase the content of VC in hemolymph, and significantly increase the expression level of key enzyme genes involved in VC synthesis and metabolism.]
Minor Comments 6: [Line 169: What RNA extraction kit was used?]
Response 6: [The reagent kit we use is RNAprep Pure Tissue Kit]
Minor Comments 7: [Line 198: What equipment was used to estimate RNA concentration and purity? If it was specrophotometer, then it is not correct to say that concentration was measutred by determining OD260/280. This ration only demonstrates nucleic acids purity from protein contamination.]
Response 7: [The purity and content of RNA were determined by determining Ultra micro UV spec-trophotometer]
Minor Comments 8: [Figure 10. The meaning of asterisks should be explained.]
Response 8: [The symbol * indicates a significant difference from the control area, while * * indicates a highly significant difference from the control area.]
Minor Comments 9: [Figure 10. In this figure, two concentrations (0.01 and 0.02) are compared, although only 0.02 mg/g was mentioned in the text.]
Response 9: [We added 0.01 mg/g experimental group in the result analysis]
Reviewer 2 Report
Comments and Suggestions for Authors
The manuscript investigates the effect of vitamin C (VC) injection on Antheraea pernyi larvae through LC-MS/MS proteomic profiling and explores the role of a putative aldolactonase gene (ApALase). The combination of proteomics, gene cloning, structural modeling, and qRT-PCR is appropriate for the research goal. However, the study is undermined by substantial issues in data analysis, presentation, figure completeness, overinterpretation of preliminary results, and lack of data transparency. A major revision is required to improve the scientific rigor, clarity, and reproducibility of the manuscript.
Major Comments
(1). The MS methods section is incomplete and lacks essential information needed for reproducibility and scientific rigor. Please revise this section to include the following:
- Quality Control (QC) Procedures
- There is no mention of any MS QC procedures (e.g., use of QC samples, system suitability checks, replicate injections, or standard peptides for retention time alignment).
- Please describe how you assessed instrument stability and run-to-run reproducibility.
- Protein and Peptide Identification Criteria
- The database used for searching (e.g., species-specific FASTA or translated transcriptome) is not clearly defined.
- Specify search parameters (e.g., enzyme specificity, number of allowed missed cleavages, variable and fixed modifications).
- Clearly state the false discovery rate (FDR) threshold applied at both the peptide-spectrum match (PSM) and protein level (you mention FDR < 1%, but do not clarify how it was controlled or estimated—e.g., target-decoy approach).
- Data Normalization and Missing Value Imputation
- There is no explanation of how protein intensities were normalized across samples (e.g., total ion current normalization, median normalization).
- State whether missing values were imputed, and if so, using what method (e.g., low abundance replacement, kNN, or model-based imputation).
- This is especially important given the small number of DEPs detected.
- Replicate Design and Statistical Testing
- Clarify how many biological and technical replicates were used per condition.
- Indicate the statistical test used for differential protein expression and whether multiple testing correction was applied (e.g., Benjamini-Hochberg adjustment).
- Data Availability
- Deposit raw MS files and processed output to a public repository such as PRIDE or MassIVE, and provide the accession number in the manuscript.
(2) Only 5 DEPs are reported, which is unusually low for shotgun proteomics.
The manuscript does not justify this result or discuss filtering criteria or data normalization methods.
A full protein table (with identifiers, fold changes, FDR, peptide counts) is needed as supplementary material.
(3) Incomplete GO/KEGG Analysis
GO/KEGG pathway analysis is limited to protein counts without enrichment statistics (e.g., p-values, FDR, enrichment scores).
Figures lack labeled axes and appropriate legends, making interpretation difficult.
Recommend reanalyzing with tools like DAVID or clusterProfiler and updating figures accordingly.
(4) Figure Clarity and Labeling
Several figures (e.g., Figures 2–6, Figure 10) lack complete axis labels, units, or explanatory legends.
Figure 10 (qRT-PCR) does not define group labels ("0", "0.01", "0.02"), error bars, statistical tests, or replicate numbers.
All figures should be self-contained and professionally formatted for clarity.
(5) Data Availability
no list of GO terms or KEGG terms, no raw qPCR data (Ct values), or qPCR efficiencies were provided.
Data availability is essential for reproducibility and must be addressed before acceptance.
(6) Overinterpretation of ApALase Function
The manuscript suggests ApALase is functionally involved in VC metabolism based on homology and expression changes.
No direct functional assays (e.g., enzymatic activity, knockdowns) are included.
Claims must be tempered, and the need for further validation clearly stated.
Minor Comments
-
Clarify what the "relative expression level" in Figure 10 is normalized to (e.g., housekeeping gene? control at 0h?).
-
Use consistent terminology: “aldolactonase” vs. “aldonolactonase” appears mixed.
-
Describe statistical methods used in figure legends (e.g., ANOVA, t-test).
-
Include full experimental parameters for RT-qPCR (primer efficiency, melt curves, reaction efficiency).
The manuscript requires significant editing to improve clarity, grammar, and fluency. There are frequent issues with sentence structure, inconsistent terminology (e.g., “aldolactonase” vs. “aldonolactonase”), and redundant phrasing—particularly in the Introduction, Discussion, and Conclusions. Some figure legends and axis labels are incomplete or ambiguous, which further impacts readability. I strongly recommend that the authors seek professional English language editing before resubmission.
Example 1
Original:
"Vitamin C is an important essential nutrient for Bombyx mori and can stimulate their feeding behavior."
Issue: Redundant wording (“important essential”); unclear antecedent of “their.”
Suggested Revision:
"Vitamin C is an essential nutrient for Bombyx mori and can stimulate its feeding behavior."
Example 2
Original:
"The analysis of differentially expressed proteins in this study was used to explore the functional annotation and classification of proteins by GO and KEGG."
Issue: Passive voice, awkward phrasing.
Suggested Revision:
"We used the differentially expressed proteins identified in this study to perform GO and KEGG functional annotation and classification."
Example 3
Original:
"The final expression results showed that ApALase gene was down-regulated with the increase of VC concentration, which may relate to its slow metabolic process."
Issue: Grammatical error (“ApALase gene was”), awkward phrasing (“may relate to”), unclear meaning.
Suggested Revision:
"The final results showed that ApALase expression decreased with increasing VC concentration, possibly due to its involvement in a slower VC metabolic process."
Author Response
Major Comments 1-1: [1. The MS methods section is incomplete and lacks essential information needed for reproducibility and scientific rigor. Please revise this section to include the following: 1-1. Quality Control (QC) Procedures. There is no mention of any MS QC procedures (e.g., use of QC samples, system suitability checks, replicate injections, or standard peptides for retention time alignment).Please describe how you assessed instrument stability and run-to-run reproducibility.]
Response 1-1: [We had added the content in the “2.2 Proteomic Analysis of Midgut Proteins in A. pernyi Larvae After VC Injection”. MaxQuant parameter configuration: MaxQuant 1.5.3.30, Enzyme (Trypsin), Pep-tide_Mass_Tolerance (4.5 ppm), Fragment_Mass_Tolerance (20 ppm), Minimal peptide length (7), PSM-level FDR (0.01), Protein-level FDR (0.01), Fixed modifications (Car-bamidomethyl (C)), Variable modifications (Oxidation (M),Acetyl (Protein N-term),Deamidated(NQ),Gln->pyro-Glu)), Database (sequence_filtered.fasta(4252 sequences)).]
Major Comments 1-2: [Protein and Peptide Identification Criteria. The database used for searching (e.g., species-specific FASTA or translated transcriptome) is not clearly defined.
Specify search parameters (e.g., enzyme specificity, number of allowed missed cleavages, variable and fixed modifications).Clearly state the false discovery rate (FDR) threshold applied at both the peptide-spectrum match (PSM) and protein level (you mention FDR < 1%, but do not clarify how it was controlled or estimated—e.g., target-decoy approach).]
Response 1-2: [We had added the content in the “2.2 Proteomic Analysis of Midgut Proteins in A. pernyi Larvae After VC Injection”. MaxQuant parameter configuration: MaxQuant 1.5.3.30, Enzyme (Trypsin), Pep-tide_Mass_Tolerance (4.5 ppm), Fragment_Mass_Tolerance (20 ppm), Minimal peptide length (7), PSM-level FDR (0.01), Protein-level FDR (0.01), Fixed modifications (Car-bamidomethyl (C)), Variable modifications (Oxidation (M),Acetyl (Protein N-term),Deamidated(NQ),Gln->pyro-Glu)), Database (sequence_filtered.fasta(4252 sequences)).]
Major Comments 1-3: [Data Normalization and Missing Value Imputation
There is no explanation of how protein intensities were normalized across samples (e.g., total ion current normalization, median normalization). State whether missing values were imputed, and if so, using what method (e.g., low abundance replacement, kNN, or model-based imputation).]
Response 1-3: [Due to the characteristics of the proteome, the number of differentially expressed proteins in this sample is indeed very small. During the analysis process, the screening threshold has been adjusted to the maximum, and the result still shows 5. However, among the only differentially expressed proteins, we found the target gene, which is in line with the research objectives of this project. Therefore, we present the results to provide assistance for future research.]
Major Comments 1-4: [Replicate Design and Statistical Testing. Clarify how many biological and technical replicates were used per condition.Indicate the statistical test used for differential protein expression and whether multiple testing correction was applied (e.g., Benjamini-Hochberg adjustment).]
Response 1-4: [Based on preliminary experiments, 4th A. pernyi larvae at 24 hours were selected for VC injection. The injection dose was 0.02 mg/g body weight, while the control group received an equal volume of 0.9% saline. Midguts from 10 larvae were collected for each sample, with 3 replicates per group. Samples were immediately frozen in liquid nitrogen and stored at -80°C for further use.]
Major Comments 1-5: [Data Availability. Deposit raw MS files and processed output to a public repository such as PRIDE or MassIVE, and provide the accession number in the manuscript.]
Response 1-5: [Thank you for your question. We are currently uploading the raw data and may need some more time. We will mark the login number at the end of the paper later.]
Major Comments 2: [Only 5 DEPs are reported, which is unusually low for shotgun proteomics.The manuscript does not justify this result or discuss filtering criteria or data normalization methods.A full protein table (with identifiers, fold changes, FDR, peptide counts) is needed as supplementary material.]
Response 2: [We had added the content in the “2.2 Proteomic Analysis of Midgut Proteins in A. pernyi Larvae After VC Injection”. Protein quantification and differential analysis: This process used MaxQuant to extract peak area intensity values and calculate protein quantification values. Then, according to the set comparison group, each Calculate the fold difference of proteins in each comparison group and perform significance testing using Welch's t-test. Fur-thermore, according to the difference factor>1.5 and P-value<0.05 as the criterion for significant differences in screening. Finally, enrichment analysis was performed on the differentially expressed proteins.
]
Major Comments 3: [Incomplete GO/KEGG Analysis. GO/KEGG pathway analysis is limited to protein counts without enrichment statistics (e.g., p-values, FDR, enrichment scores).Figures lack labeled axes and appropriate legends, making interpretation difficult. Recommend reanalyzing with tools like DAVID or clusterProfiler and updating figures accordingly.]
Response 3: [We had added the content in the “3.1 Proteomic Data of Midgut Proteins in A. pernyi Larvae After VC Injection and GO Functional Annotation”. The mass spectrometry instrument used in this study was the TripleTOF 5600. The raw data were processed using MaxQuant with the Andromeda search engine. Data were filtered at the PSM-level FDR ≤1% and further filtered at the Protein-level FDR ≤1%. A total of 166 proteins and 1,858 peptides were identified across all samples]
Major Comments 4: [Figure Clarity and Labeling. Several figures (e.g., Figures 2–6, Figure 10) lack complete axis labels, units, or explanatory legends. Figure 10 (qRT-PCR) does not define group labels ("0", "0.01", "0.02"), error bars, statistical tests, or replicate numbers. All figures should be self-contained and professionally formatted for clarity.]
Response 4: [Thank you very much for your correction. We have made modifications to the images in the article.]
Major Comments 5: [Data Availability no list of GO terms or KEGG terms, no raw qPCR data (Ct values), or qPCR efficiencies were provided. Data availability is essential for reproducibility and must be addressed before acceptance.]
Response 5: [Thank you for your question. We had added the content in the “2.2 Proteomic Analysis of Midgut Proteins in A. pernyi Larvae After VC Injection”. MaxQuant parameter configuration: MaxQuant 1.5.3.30, Enzyme (Trypsin), Pep-tide_Mass_Tolerance (4.5 ppm), Fragment_Mass_Tolerance (20 ppm), Minimal peptide length (7), PSM-level FDR (0.01), Protein-level FDR (0.01), Fixed modifications (Car-bamidomethyl (C)), Variable modifications (Oxidation (M),Acetyl (Protein N-term),Deamidated(NQ),Gln->pyro-Glu)), Database (sequence_filtered.fasta(4252 sequences)).]
Major Comments 6: [Overinterpretation of ApALase Function. The manuscript suggests ApALase is functionally involved in VC metabolism based on homology and expression changes. No direct functional assays (e.g., enzymatic activity, knockdowns) are included. Claims must be tempered, and the need for further validation clearly stated.]
Response 6: [Thank you for your question. In the “5. Conclusions”, we have added “In the next step of evolutionary research, we will continue to investigate the function of this gene by measuring the enzyme activity of key enzymes in the VC metabolic pathway under treatment with different concentrations of VC injection, and validating the gene function from physiological and biochemical perspectives. If possible, conduct knockout experiments on this gene in oak silkworms to observe its function more intuitively.”
]
Minor Comments 1: [Clarify what the "relative expression level" in Figure 10 is normalized to (e.g., housekeeping gene? control at 0h?).]
Response 1: [Thank you very much for your correction. We have made modifications to the images in the article.]
Minor Comments 2: [Use consistent terminology: “aldolactonase” vs. “aldonolactonase” appears mixed.]
Response 2: [Thank you very much for your correction. The title of the article has been revised and the article will be uniformly used“aldonolactonase”.]
Minor Comments 3: [Describe statistical methods used in figure legends (e.g., ANOVA, t-test).]
Response 3: [Thank you for your correction. As the contribution to the overall experimental purpose is not significant, the section on phylogenetic tree has been deleted.]
Minor Comments 4: [Include full experimental parameters for RT-qPCR (primer efficiency, melt curves, reaction efficiency).]
Response 4: [Due to the different formats of the instruments used in the experiment, a regular RT-qPCR program that can be opened was not saved. Subsequently, the file in ".xlsx" format was transferred to the original file to prove that the RT-qPCR experiment was conducted. In future experiments, special attention will be paid to such issues. We hope the teacher can understand]
Comments on the Quality of English Language
The manuscript requires significant editing to improve clarity, grammar, and fluency. There are frequent issues with sentence structure, inconsistent terminology (e.g., “aldolactonase” vs. “aldonolactonase”), and redundant phrasing—particularly in the Introduction, Discussion, and Conclusions. Some figure legends and axis labels are incomplete or ambiguous, which further impacts readability. I strongly recommend that the authors seek professional English language editing before resubmission.
Response: [Thank you for your criticism and correction. The manuscript has been submitted to a professional English editor for revision. Please review it again.]
Reviewer 3 Report
Comments and Suggestions for Authors
Summary:
In this article, the authors have explored the mechanism of vitamin C metabolism in Antheraea pernyi. They have compared and examined the proteome of the midgut of tussah larvae before and after injection of VC and identified and cloned the Antheraea pernyi aldolactonase gene related to the metabolism of VC in A. pernyi. Their differential proteomic data revealed that catalytic activity is a protein implicated in the metabolism of key enzymes. The introduction is relevant and theory based. Sufficient details about the previous research findings are shown for readers. However, there are some comments that need to be addressed.
Major comments
- It is recommended to change “VC” to vitamin C since the title of the article should be precise and avoid abbreviations to assure better understanding and accessibility for the reader.
- The last phrase of the introduction section should be extended by briefly mentioning the major aim of the work and highlighting the principal conclusions.
- It is highly recommended to define the abbreviations for the first time used. Further, there are various abbreviations in the manuscript without definitions. Therefore, it is recommended to define all of them.
- In the article the term “in vitro” should be written in italic font.
- The discussion section is short and misses the limitations. Thus, the authors should extend it by mentioning other previous recent studies and including some limitations of their research and how they can improve it in their future study.
- The conclusion section is so long. It should be short and summarize the main results of the study without references.
Author Response
Comments 1: [It is recommended to change “VC” to vitamin C since the title of the article should be precise and avoid abbreviations to assure better understanding and accessibility for the reader.]
Response 1: [Thank you for your correction. We have already made modifications in the article.]
Comments 2: [The last phrase of the introduction section should be extended by briefly mentioning the major aim of the work and highlighting the principal conclusions.]
Response 2: [Thank you for your correction. The article has been revised accordingly.]
Comments 3: [It is highly recommended to define the abbreviations for the first time used. Further, there are various abbreviations in the manuscript without definitions. Therefore, it is recommended to define all of them.]
Response 3: [Thank you for your question. The article has been revised accordingly.]
Comments 4: [In the article the term “in vitro” should be written in italic font.]
Response 4: [Thank you for your correction. We have already made modifications in the article.]
Comments 5: [The discussion section is short and misses the limitations. Thus, the authors should extend it by mentioning other previous recent studies and including some limitations of their research and how they can improve it in their future study.]
Response 5: [Thank you for your question. The article has been revised accordingly.]
Comments 6: [The conclusion section is so long. It should be short and summarize the main results of the study without references.]
Response 6: [Thank you for your correction. We have already made modifications in the article.]
Round 2
Reviewer 1 Report
Comments and Suggestions for Authors
In the revised version of the manuscript, the authors took into account the majority of the comments made for the initial version. At the same time, there are some points taht still need to be improved before acceptance:
- Lines 41-51: The reference numbers should be modofoed according to their sequence in the text.
- Lines 89-92: These two sentences should be moved above in the text, because they are not appropriate to finish the Introduction.
- Section 2.2. I do not like the way the authors describe the proteomic analysis. First, each paragraph ('protein extraction', 'HPLC', Mass spectiometry',.....) could be indicated as 2.2.1., 2.2.2., etc. Also, the methods description should be as follows: 'An appropriate amount of sample was transferred,....DTT was added, .... etc. Now the description looks more as an instruction, not scientific paper.
- Line 235: What commercial kit was used for RT?
- Line 256: This information should be in Materials and Methods, not Results.
- Lines 432-451: This paragraph should be moved to Discussion. In general, such a short Discussion and lengthy Conclusions look quite strange.
Author Response
Comments 1: [Lines 41-51: The reference numbers should be modofoed according to their sequence in the text.]
Response 1: [Thank you for your correction. The article has been revised accordingly.]
Comments 2: [Lines 89-92: These two sentences should be moved above in the text, because they are not appropriate to finish the Introduction.]
Response 2: [Thank you for your correction. These two sentences have been moved to the top of the main text.]
Comments 3: [Section 2.2. I do not like the way the authors describe the proteomic analysis. First, each paragraph ('protein extraction', 'HPLC', Mass spectiometry',.....) could be indicated as 2.2.1., 2.2.2., etc. Also, the methods description should be as follows: 'An appropriate amount of sample was transferred,....DTT was added, .... etc. Now the description looks more as an instruction, not scientific paper.]
Response 3: [Thank you for your correction. We have divided 2.2 into sections and made revisions to the entire description, which conforms to the normal writing format of the paper. Please review again.]
Comments 4: [Line 235: What commercial kit was used for RT?]
Response 4: [The commercial kit of RT was Takara Universal SYBR qPCR Master Mix PCR. We have added explanations in the article, thank you for your correction. ]
Comments 5: [Line 256: This information should be in Materials and Methods, not Results.]
Response 5: [This content is in the materials and methods section, not in the results section. Please verify again.]
Comments 6: [Lines 432-451: This paragraph should be moved to Discussion. In general, such a short Discussion and lengthy Conclusions look quite strange.]
Response 6: [We have made adjustments to the conclusion and discussion sections. The conclusion is simple and clear, and the discussion is quite rich. Thank you for your correction, and please review it again]
Reviewer 2 Report
Comments and Suggestions for Authors
I thank the authors for their revisions and detailed responses to my earlier review. I appreciate the effort made in addressing the various comments. However, after carefully evaluating the responses and the revised text (as described in the responses), I find that several of the major points have not been fully addressed in a satisfactory manner, particularly those concerning methodological transparency, reproducibility, and data availability.
Below I provide point-by-point feedback on your responses.
Major Comment 1-1: QC Procedures
Reviewer’s original request: Please describe QC procedures for MS runs (e.g., use of QC samples, system suitability checks, replicate injections, retention time alignment).
Author’s response: The response simply repeats MaxQuant search parameters, which are unrelated to QC or instrument stability checks.
Assessment: Not adequately addressed.
Recommendation:
- The authors must explicitly describe any system suitability checks (e.g., standard peptide or digest injections).
- Indicate if pooled QC samples or replicates were used to monitor stability across the run.
- If no QC procedures were performed, please state this explicitly and discuss as a limitation.
Major Comment 1-2: Protein and Peptide Identification Criteria
Reviewer’s original request: Clarify the database origin (species-specific?), enzyme specificity, missed cleavages, modifications, FDR thresholds, and how FDR was controlled (target-decoy approach).
Author’s response: Provided MaxQuant parameters, including FDR thresholds and modifications, but still unclear about database origin.
Assessment: Partially addressed.
Recommendation:
- Clarify the origin and construction of the FASTA database. Was it species-specific? Was it a translated transcriptome or curated proteome?
- Explicitly confirm use of a target-decoy approach for FDR estimation.
Major Comment 1-3: Data Normalization and Missing Value Imputation
Reviewer’s original request: Explain normalization across samples and imputation strategy.
Author’s response: Did not address either point.
Assessment: Not addressed at all.
Recommendation:
- Specify the normalization method (e.g., MaxLFQ algorithm, total ion current).
- Describe whether missing values were imputed and which method was used (e.g., normal distribution-based imputation in Perseus, kNN).
Major Comment 1-4: Replicate Design and Statistical Testing
Reviewer’s original request: Clarify biological/technical replicates, statistical tests used, and multiple testing correction.
Author’s response: Stated biological replicates (n=3) but no mention of statistical test or multiple testing correction.
Assessment: Partially addressed.
Recommendation:
- Please state the exact statistical test used (e.g., Welch’s t-test).
- Clearly indicate whether multiple testing correction (e.g., Benjamini-Hochberg) was applied to control FDR in differential expression analysis.
Major Comment 1-5: Data Availability
Reviewer’s original request: Deposit raw MS data to PRIDE/MassIVE with accession number.
Author’s response: Said they are uploading but will add accession later.
Assessment: Acceptable for now, assuming they will ensure it is added before acceptance.
Recommendation:
- Please commit to providing the accession number in the final accepted version.
Major Comment 2: Only 5 DEPs Reported
Reviewer’s original request: Justify low number of DEPs, describe filtering criteria, provide full protein table as supplement.
Author’s response: Claimed few DEPs are consistent with project goals but did not include any table or detailed thresholds.
Assessment: Insufficient.
Recommendation:
- Authors must add a full supplementary table with all quantified proteins, UniProt IDs, fold changes, p-values, FDR, and peptide counts.
- Clearly define the fold-change and significance thresholds used for filtering.
Major Comment 3: Incomplete GO/KEGG Analysis
Reviewer’s original request: Provide enrichment statistics (p-values, FDR), improve figure clarity.
Author’s response: Added total number of proteins/peptides but did not mention enrichment p-values or FDR.
Assessment: Not adequately addressed.
Recommendation:
- Please reanalyze GO/KEGG enrichment using standard tools (e.g., DAVID, clusterProfiler).
- Provide enrichment p-values, FDR-corrected values, and enrichment scores.
- Include complete lists as supplementary tables.
Major Comment 5: Data Availability (GO/KEGG terms, qPCR raw data)
Reviewer’s original request: Provide lists of GO/KEGG terms and raw qPCR data.
Author’s response: Did not provide lists or raw data—only repeated MaxQuant parameters.
Assessment: Not addressed at all.
Recommendation:
- Upload complete lists of enriched GO and KEGG terms.
- Include raw qPCR Ct values, amplification efficiencies, melt curves in the Supplementary Materials.
Minor Comments
Minor 3: Response unclear (deleted phylogenetic tree section instead of describing statistics in figure legends).
Recommendation:
- Ensure all figure legends include the statistical test used.
Minor 4: Explanation given about lost qPCR files, with promise to improve future record-keeping.
Acceptable as a limitation note.
Author Response
Major Comments 1-1: [1. The MS methods section is incomplete and lacks essential information needed for reproducibility and scientific rigor. Please revise this section to include the following: 1-1. Quality Control (QC) Procedures. There is no mention of any MS QC procedures (e.g., use of QC samples, system suitability checks, replicate injections, or standard peptides for retention time alignment).Please describe how you assessed instrument stability and run-to-run reproducibility.]
Response 1-1: [MaxQuant( http://www.maxquant.org) is a free protein identification and quanti-fication software developed by Max Planck Institutes in Germany, suitable for high-precision mass spectrometry data. The software version used in this project is MaxQuant 1.5.3.30. When operating, use the original offline data as the input file, con-figure the corresponding parameters and database, and then perform identification and quantitative analysis.
I have already added it to the article of 2.2.6, thank you for your correction.]
Major Comments 1-2: [Protein and Peptide Identification Criteria. The database used for searching (e.g., species-specific FASTA or translated transcriptome) is not clearly defined.
Specify search parameters (e.g., enzyme specificity, number of allowed missed cleavages, variable and fixed modifications).Clearly state the false discovery rate (FDR) threshold applied at both the peptide-spectrum match (PSM) and protein level (you mention FDR < 1%, but do not clarify how it was controlled or estimated—e.g., target-decoy approach).]
Response 1-2: [The enzyme specificity is trypsin.
Variable and fixed modifications Variable modifications is Oxidation (M),Acetyl (Protein N-term),Deamidated(NQ),Gln->pyro-Glu, and fixed modifications is fixed modifications Carbamidomethyl (C).
The mass spectrometry instrument used in this study was the TripleTOF 5600. The raw data were processed using MaxQuant with the Andromeda search engine. Data were filtered at the PSM-level FDR ≤1% and further filtered at the Protein-level FDR ≤1%.]
Major Comments 1-3: [Data Normalization and Missing Value Imputation
There is no explanation of how protein intensities were normalized across samples (e.g., total ion current normalization, median normalization). State whether missing values were imputed, and if so, using what method (e.g., low abundance replacement, kNN, or model-based imputation).]
Response 1-3: [This process uses MaxQuant to extract peak area intensity values and calculate protein quantification values. Then, according to the set comparison group, each Calculate the fold difference of proteins in each comparison group and perform significance testing using Welch's t-test. Furthermore, according to the difference factor>1.5. Use P-value<0.05 as the criterion for significant differences in screening. Finally, enrichment analysis was performed on the differentially expressed proteins.
I have already added it to the article of 2.2.8, thank you for your correction.]
Major Comments 1-4: [Replicate Design and Statistical Testing. Clarify how many biological and technical replicates were used per condition.Indicate the statistical test used for differential protein expression and whether multiple testing correction was applied (e.g., Benjamini-Hochberg adjustment).]
Response 1-4: [The offline data was evaluated using the Andromeda engine integrated with MaxQuant, and at the spectral level using PSM level FDR<=1% completion of filtration, further filtration at protein level with FDR<=1% to obtain significant identification results.
I have already added it to the article of 2.2.8, thank you for your correction.]
Major Comments 1-5: [Data Availability. Deposit raw MS files and processed output to a public repository such as PRIDE or MassIVE, and provide the accession number in the manuscript.]
Response 1-5: [The accession number in the manuscript is IPX0012574000.]
Major Comments 2: [Only 5 DEPs are reported, which is unusually low for shotgun proteomics.The manuscript does not justify this result or discuss filtering criteria or data normalization methods.A full protein table (with identifiers, fold changes, FDR, peptide counts) is needed as supplementary material.]
Response 2: [The specific identification results of each sample are shown in the article of 3.1, described from three aspects: "unique peptide distribution", "protein mass distribution", and "protein coverage distribution"]
Major Comments 3: [Incomplete GO/KEGG Analysis. GO/KEGG pathway analysis is limited to protein counts without enrichment statistics (e.g., p-values, FDR, enrichment scores).Figures lack labeled axes and appropriate legends, making interpretation difficult. Recommend reanalyzing with tools like DAVID or clusterProfiler and updating figures accordingly.]
Response 3: [For proteins that are significantly down regulated, GO enrichment analysis provides GO entries for differentially enriched proteins, which often involve biological functions that researchers are most concerned about. This result usually uses P-value<0.05 as a significantly enriched GO entry.
The following screenshot shows the results of the Pathway enrichment analysis webpage. Among them, pathways with Pvalue<0.05 are significantly enriched in differentially expressed proteins.
I have already added it to the article of 3.1 and 3.2, thank you for your correction.]
Major Comments 5: [Data Availability no list of GO terms or KEGG terms, no raw qPCR data (Ct values), or qPCR efficiencies were provided. Data availability is essential for reproducibility and must be addressed before acceptance.]
Response 5: [Ontology GO: The category of the ontology (biology_process, cellar_component, or molecular_function); Class: GO entry; Number of *: proteins annotated to each GO entry using 'numb_of_ annotate; Proteins_of_ *: to the protein IDs of each GO entry.
Code: COG or KOG functional code; Functional-Categories: COG or KOG Functional Categories; Protein-Number: The number of proteins classified by various COG or KOG functions Sequence IDs of Proteins in Functional Categories of COG or KOG
Protein: The name of a protein sequence; Protein-or-Domain: COG or KOG annotation alignment of proteins or domains. Score: COG or KOG annotation Blast comparison score;
E-Value: COG or KOG annotation Blast comparison of E-Value; COG/KOG-ID: The ID of the COG or KOG annotation; Function-Description: The functional description of COG or KOG annotations; Functional classification code for COG or KOG annotations; Functional Categories: COG or KOG annotated functional categories.]
Reviewer 3 Report
Comments and Suggestions for Authors
The changes introduced by the authors are acceptable, considering some of my comments. However, regarding improvements to the discussion and conclusion sections, they are not acceptable. The discussion remains brief and fails to mention limitations, and the conclusion section appears to be more of a discussion than a conclusion. As I mentioned earlier, the conclusion should be brief and summarize the main results of the study without references.
Author Response
Comments 1: [The changes introduced by the authors are acceptable, considering some of my comments. However, regarding improvements to the discussion and conclusion sections, they are not acceptable. The discussion remains brief and fails to mention limitations, and the conclusion section appears to be more of a discussion than a conclusion. As I mentioned earlier, the conclusion should be brief and summarize the main results of the study without references.]
Response 1: [We have made adjustments to the conclusion and discussion sections. The conclusion is simple and clear, and the discussion is quite rich. Thank you for your correction, and please review it again.]
Round 3
Reviewer 2 Report
Comments and Suggestions for Authors
Thanks authors for the response and correction. However, there are still several questions which needs the clarification.
Major concern 1-1, I mainly concern about the stability of Mass spectrometer during the measurement of your samples. I'd like to make sure the machine runs on a stable and consistent performance. It is not what you answered which kind of software you used for MS data decipher. If there is no quality control, please be honest to answer no. But because you used the abundance information for comparsion as the lable-free quantitation. It is related to the confidence of identified abundance of all proteins across different samples. This is aslo linked to my mahor comment 1-3. whethe you normalize the orginal peak area across 6 samples prior to statistically significant analysis.
Major 1-2, the databse is unclear to readers. You only mention that sequence-filtered. fasta (4252 sequences). I think this databse file is created by youself based on genomic data, right? Because we can not download it from Uniport, or any other open databse. You have to upload it for the reader to download and allow the reader to re-analyze the raw files using this database file. Please add it as a supplementary file.
Major comment 3. only show the number of proteins in each GO entry is not enough. You have to provide the FDR or P-value show which proteins are over-representation or down-representation. Eg, string, DAVAID, Cytoscrape, GO clusterfile. Any of this softwares enable to do it.
Author Response
Major concern 1-1: [I mainly concern about the stability of Mass spectrometer during the measurement of your samples. I'd like to make sure the machine runs on a stable and consistent performance. It is not what you answered which kind of software you used for MS data decipher. If there is no quality control, please be honest to answer no. But because you used the abundance information for comparsion as the lable-free quantitation. It is related to the confidence of identified abundance of all proteins across different samples. This is aslo linked to my mahor comment 1-3. whethe you normalize the orginal peak area across 6 samples prior to statistically significant analysis.]
Response 1-1: [After using Maxquant analysis on the sample data, quantitative values of each protein were obtained, and then analyzed separately from the "intra group coefficient of variation" and "principal component analysis"
Evaluate the quality of data in terms of sample quantitative correlation and other aspects.]
Major 1-2: [the databse is unclear to readers. You only mention that sequence-filtered. fasta (4252 sequences). I think this databse file is created by youself based on genomic data, right? Because we can not download it from Uniport, or any other open databse. You have to upload it for the reader to download and allow the reader to re-analyze the raw files using this database file. Please add it as a supplementary file.]
Response 1-2: [The website for uploading data can be downloaded directly.]
Major comment 3. only show the number of proteins in each GO entry is not enough. You have to provide the FDR or P-value show which proteins are over-representation or down-representation. Eg, string, DAVAID, Cytoscrape, GO clusterfile. Any of this softwares enable to do it.
Response 1-2: [Supplementary figures 4 and 6 have been added to the results section of the article. Please review them.]